# Influence of Dynamic Strain Sweep on the Degradation Behavior of FeMnSi–Ag Shape Memory Alloys

**DOI:** 10.3390/jfb14070377

**Published:** 2023-07-19

**Authors:** Ana-Maria Roman, Ramona Cimpoeșu, Bogdan Pricop, Nicoleta-Monica Lohan, Marius Mihai Cazacu, Leandru-Gheorghe Bujoreanu, Cătălin Panaghie, Georgeta Zegan, Nicanor Cimpoeșu, Alice Mirela Murariu

**Affiliations:** 1Faculty of Materials Science and Engineering, “Gheorghe Asachi” Technical University of Iași, Blvd. Dimitrie Mangeron 71A, 700050 Iași, Romania; ana-maria.roman@academic.tuiasi.ro (A.-M.R.); ramona.cimpoesu@academic.tuiasi.ro (R.C.); bogdan.pricop@academic.tuiasi.ro (B.P.); nicoleta-monica.lohan@academic.tuiasi.ro (N.-M.L.); leandru-gheorghe.bujoreanu@academic.tuiasi.ro (L.-G.B.); catalin.panaghie@student.tuiasi.ro (C.P.); 2Physics Department, “Gheorghe Asachi” Technical University of Iași, Blvd. Dimitrie Mangeron 71A, 700050 Iași, Romania; marius-mihai.cazacu@academic.tuiasi.ro; 3Faculty of Dental Medicine, “Grigore T. Popa” University of Medicine and Pharmacy University, 16 University Street, 700115 Iasi, Romania; 4Department of Surgicals, Faculty of Dental Medicine, “Grigore T. Popa” University of Medicine and Pharmacy, 16 University Street, 700115 Iasi, Romania; alice.murariu@umfiasi.ro

**Keywords:** FeMnSi–Ag biodegradable alloy, SMAs, SEM, EDS, AFM, DMA, DSC, FTIR and nano-FTIR

## Abstract

Iron-based SMAs can be used in the medical field for both their shape memory effect (SME) and biodegradability after a specific period, solving complicated chirurgical problems that are partially now addressed with shape-memory polymers or biodegradable polymers. Iron-based materials with (28–32 wt %) Mn and (4–6 wt %) Si with the addition of 1 and 2 wt % Ag were obtained using levitation induction melting equipment. Addition of silver to the FeMnSi alloy was proposed in order to enhance its antiseptic property. Structural and chemical composition analyses of the newly obtained alloys were performed by X-ray diffraction (confirming the presence of ε phase), scanning electron microscopy (SEM) and energy-dispersive spectroscopy. The corrosion resistance was evaluated through immersion tests and electrolyte pH solution variation. Dynamic mechanical solicitations were performed with amplitude sweep performed on the FeMnSi–1Ag and FeMnSi–2Ag samples, including five deformation cycles at 40 °C, with a frequency of 1 Hz, 5 Hz and 20 Hz. These experiments were meant to simulate the usual behavior of some metallic implants subjected to repetitive mechanical loading. Atomic force microscopy was used to analyze the surface roughness before and after the dynamic mechanical analysis test followed by the characterization of the surface profile change by varying dynamic mechanical stress. Differential scanning calorimetry was performed in order to analyze the thermal behavior of the material in the range of −50–+200 °C. X-ray diffraction and Fourier transform infrared spectroscopy (FTIR) along with Neaspec nano-FTIR experiments were performed to identify and confirm the corrosion compounds (oxides, hydroxides or carbonates) formed on the surface.

## 1. Introduction

The field of biodegradable alloys has been a very common theme in scientific studies over time. Researchers have addressed many directions in the study of biocompatibility associated with mechanical properties and corrosion rates to reach the right combination needed in the healing process of diseased tissues [1]. Temporary biodegradable implants used in medical applications must provide mechanical support for a certain period during healing and degrade after the tissue has been remodeled [2].

Fe–Mn-based alloys have been studied under different aspects, showing good biocompatibility properties [3], mechanical strength [4] and magnetic resonance imaging (MRI) compatibility, i.e., non-magnetism due to the austenitic phase [5]. Alloying with Si has led to shape memory properties very useful in the design of potential medical implants, such as stents for cardiological applications [6], and increased corrosion rate [7]. Mn and Si are non-toxic and essential elements in the mammalian body responsible for its proper functioning [8,9,10,11]. Fe–Mn–Si alloys with Mn content between 25 and 30 wt % possess mechanical properties close to those of stainless steel [6,12,13] and the biomechanical compatibility required for bone implants [14,15].

Shape memory alloys (SMAs) are well known for their properties such as superelasticity and the ability to recover a memorized shape upon the action of a stimulus [16]. The best known biocompatible SMAs are NiTi-based ones [17] with excellent mechanical properties, used as permanent implants. What is related to this type of implant are the medical complications that may occur in the long term, requiring additional surgery [18]. In some applications where the temporary presence of non-degradable biocompatible alloys is required, further surgery may also be required for removal, which could affect the patient’s health [19]. The design of biodegradable SMAs would lead to facilitating the healing process, providing mobility in some specific applications, remodeling and healing the diseased tissue so that no further surgery is required afterward [20,21]. Based on its excellent formability, low manufacturing cost, biocompatibility and biodegradability in physiological environment as reported by in vitro studies investigating Fe–Mn–Si alloy as SMA as well as its biodegradability in physiological environment, a biodegradable Fe-based SMA could be a suitable candidate for temporary implants [22,23,24,25,26].

Due to the rather low corrosion rates of Fe-based alloys, which are intended to be improved [27,28], the technique of alloying with noble elements such as Au, Pt, Pd and Ag has been used, which has led to increased corrosion rates through micro-galvanic corrosion due to the potential differences observed between the matrix that acted as the anode and phases containing precipitates of the noble metals [29,30,31].

The choice of Ag for alloying was due to its well-known antimicrobial properties [32]. Ag is used in medical devices as an antibiotic, interacting with the human body [33]. Studies indicate that Ag absorbed by the human body is eliminated through natural pathways by the kidneys and liver [34]. By alloying Fe with Ag, a new biodegradable alloy was produced for further studies, thus it was reported that a microstructure was formed in which Ag particles are dispersed in the Fe matrix, as Ag is known to be immiscible in Fe [35]. Niendorf et al. [36] observed that, in the TWIP–5Ag alloy (Fe–Mn-based), due to the potential difference, the Ag phase and the Fe matrix formed a galvanic couple leading to increased degradation rate and insignificant changes in mechanical properties compared to the basic Ag-free TWIP alloy. In the experiments conducted on biodegradable Fe–Mn–Ag alloys, Liu et al. [37] reported for Fe70.3–28.9Mn–0.8Ag alloy lower magnetic susceptibility and an increase in the corrosion rate compared to Fe–30Mn without Ag content.

Tonna et al. [38] studied Fe, Fe35Mn and (Fe35Mn)5Ag samples in Hanks’ Balanced Salt Solution (HBSS) with and without Ca^2+^ and reported that (Fe35Mn)5Ag had the highest corrosion current density in HBSS containing Ca^2+^, concluding that the addition of Ag leads to an accelerated corrosion rate in the electrolyte that is most representative for orthopedic products. In the second part of the same study, Wang et al. [39] performed corrosion tests. They reported that the addition of 5 wt % Ag to the FeMn alloy created micro-galvanic couplings between the austenitic FeMn matrix and the Ag phase. These couplings served as effective cathodic sites for the oxygen reduction reaction (ORR) that produced quantities of OH, favoring the deposition of Ca–P products on the metal surface that stabilized the local pH.

Sotoudeh Bagha et al. [40] used mechanical alloying and Spark Plasma Sintering to obtain different microstructures, namely, nano, macro and bimodal in Fe–30Mn–(1-3)Ag alloys. The results showed better mechanical properties and a higher corrosion rate (0.88 mm/yr) for the bimodal Fe–30Mn–1Ag alloy [40]. In another study, Sotoudeh Bagha et al. [41] obtained promising results for the alloys with Ag addition compared to Fe–Mn alloy without Ag content, regarding increased hardness and ultimate shear strength of Fe-30Mn–1Ag (156 HV, 360 MPa) and Fe–30Mn–3Ag (174 HV, 490 MPa). In addition, Fe-30Mn–1Ag showed promising antibacterial activity against *Staphylococcus aureus* (*S. aureus*) and *Escherichia coli* (*E. coli*) bacteria.

Babacan et al. [42] conducted a study on Fe–30Mn–6Si, Fe30Mn–6Si–0.6Ag and Fe–30Mn–6Si–1.2Ag (wt %) alloys, where they reported higher mechanical strength values and faster degradation in simulated body fluid (SBF) than 316 L stainless steel and Fe–30Mn alloy. Increasing Ag content led to higher hardness which was explained by the increased amount of observed ε hexagonal close-packed (hcp)-martensite.

In vitro experiments are performed by immersing samples in physiological solutions, which are more or less aggressive aqueous media (e.g., Ringer’s solution, Hank’s solution, Simulated Body Fluid (SBF)) [4,6,10,23]. The corrosion mechanism of titanium dioxide nanotubes in Ringer’s solution was studied by Zarebidaki [43] and Chelliah studied the evaluation of electrochemical impedance and biocorrosion characteristics of as-cast and T4 heat-treated AZ91 Mg alloys in Ringer’s solution [44]. To preserve the mechanical integrity of the implant, damage should start extremely slowly and then increase at the same rate as the body’s natural healing process. It takes 6 to 12 months to complete the remodeling procedure when significant medical problems are involved. It is essential to note that iron degradation should not occur at a rate that leads to excessive deposits of degradation products around the implantation site.

Kinetic factors influence corrosion rates, while thermodynamic aspects influence corrosion behavior [45]. Chloride ions found in physiological fluids, such as blood and interstitial fluid, play a major role in determining the degree of metal corrosion in living organisms [46]. In actual circumstances, a human body fluid’s pH varies slightly and serves as a buffer solution. A pH of 7.3 to 7.45 is considered normal for blood and interstitial fluids, while it might reduce in close proximity to protein isoelectric points and surface implantation sites [46]. For porous iron created with additives in SBF solution, Sharma et al. [47] examined the pH increase during a 28-day period. The results established a pH range of 0.50 to 0.05 in the sample.

The degree of corrosion kinetics and the degradation process are influenced by specific characteristics, including surface film state and environmental factors (such as pH and flow rate). The Pourbaix diagram and Pilling–Bedworth ratio both show these aspects [48]. The partial anodic reaction (metal dissolution) is the first reaction that occurs upon contact of the metal with the physiological medium (Equations (1) and (2)) [49]. Performing in vitro immersion tests in Hank’s solution, Dargusch et al. [50] demonstrated that Mn in the Fe layer appears in two valence states, Mn^+2^ and Mn^+3^, as demonstrated by XPS investigation. Mn(OH)_2_, Mn(OOH), and Mn_2_O_3_ were found to be present. Mn can be oxidized through reaction (2) when Fe is present.
Fe → Fe^+2^ + 2e^−^,(1)
Mn → Mn^+2^ + 2e^−^,(2)

When iron corrodes, the rate is usually controlled by the cathodic oxygen reduction reaction (Equation (3)) [51]:2H_2_O + O_2_ + 4e^−^ → 4OH^−^,(3)

Iron hydroxide (Fe(OH)_2_) or hydrating iron oxide (FeO·nH_2_O) are produced when metal ions are released and react with hydroxyl ions (Equation (4)) [52]:Fe^+2^ + 2OH^−^ → Fe(OH)_2_,(4)

FeO·nH_2_O or Fe(OH)_2_ are always in the adjacent layer to the metal interface when corrosion occurs. Ferrous oxides (Fe^2+^) are subsequently transformed to hydrous ferric oxide (Fe^3+^) or ferric hydroxide at the outside of the hydroxide layer due to the presence of dissolved oxygen [51], as shown in Equation (5):Fe(OH)_2_ + 1/2H_2_O + 1/4O_2_ → Fe(OH)_3_,(5)

Fe hydroxides may also take the Fe oxide forms of hematite (Fe_2_O_3_), magnetite (Fe_3_O_4_), and wustite (FeO), according to Hermawan et al.’s research [52]. They also claimed that corrosion pitting might occur when chloride ions (Cl^−^) react with iron ions through an autocatalytic mechanism (Equations (6) and (7)) [52]:Fe^+2^ + 2Cl^−^ → FeCl_2_(6)
FeCl_2_ + 2H_2_O → Fe(OH)_2_ + 2HCl(7)

This work aims to obtain and characterize two FeMnSi–Ag SMAs as biodegradable alloys. Isothermal dynamic strain sweeps will be performed and their influence on the corrosion resistance will be investigated. Macro and nano investigations will be performed on corrosion compounds formed after immersion in Ringer’s solution.

## 2. Materials and Methods

Two experimental Fe29Mn5Si1Ag and Fe29Mn5Si2Ag (wt %) SMAs were obtained from high-purity materials (99.98% electrolytic Fe, 99.7% electrolytic metallic Mn, 99.5% metallic Si, 999.6 ‰ fine Ag, by melting and remelting the first ingots in magnetic levitation induction furnace with cold crucible Fives Celes (Lautenbach, France) at R&D Consultancy & Services, Bucharest under Ar atmosphere. These specimens will be further designated as FeMnSi–1Ag and FeMnSi–2Ag. The ingots resulting from the re-melting were processed by cutting and turning, and finally they had a diameter of 18.5 mm and a height of 35–40 mm. Cylindrical samples of 3–4 mm thickness and parallelepiped samples of approximately 25 × 15 × 3 mm were prepared. The parallelepiped specimens were subjected to hot rolling at 1050 °C during three passes until a thickness of 1 mm was obtained.

X-ray diffraction (XRD) tests were made for phase analysis with an Expert PRO-MPD system (Panalytical, Almelo, The Netherlands type, Cu-X-ray tube (Kα-1.54°)). The prepared samples measuring 25 × 4 × 1 mm, were grinded on metallographic paper up to 5000 grit and cleaned with technical alcohol in an ultrasonic bath (PRO 50 ASonic, ULTRASONIC CLEANER, Shenzhen, China) for 60 min.

Differential scanning calorimetry (DSC) experiments were performed on two sample fragments (weighing approximately 40 mg) from each state (hot-rolled and cast) after the cutting procedure and were subsequently cleaned from oxides in an ultrasonic bath with technical alcohol for 60 min. The samples were weighed with a digital laboratory balance (PARTNER AS 220/C/2, RADWAG Balances & Scales, Radom, Poland). NETZSCH DSC equipment model DSC 200 F3 Maya (Netzsch, Selb, Germany, with sensitivity less than l W, temperature variation of 0.1 K and enthalpy accuracy generally <1%) was used for the experiments. Calibration of the equipment was performed with reference materials of Hg, Bi, In, Sn and Zn. Experimental setup of the DSC: one cooling cycle from room temperature (RT) ~25 °C to −50 °C and one heating cycle from −50 °C to 200 °C and again cooling to RT. The cooling and heating rate of the experiment was 10 K/min and was performed under protective atmosphere of Ar. For the analysis of the thermal behavior of the sample during cooling and heating, the Proteus software provided by NETZSCH was used using the tangent method.

For the dynamic mechanical analysis (DMA), samples cut with an electro-discharge machine (EDM) were used, with the following dimensions: 25 × 4 × 1 mm that were prepared by grinding on metallographic paper up to 5000 grit and cleaned in technical alcohol for 60 min. For each individual alloy, samples were tested by strain sweeps performed with a three-point-bending specimen holder (DMA-SS-3PB) using an Artemis NETZSCH (Netzsch, Selb, Germany) model DMA 242 analyzer. The equipment has a force resolution of 5 × 10^−3^ N, amplitude range: ±0.1 to 240 μm and amplitude resolution: 5 × 10^−3^. DMA-SS-3PB performed on the FeMnSi–1Ag sample comprised 5 deformation cycles at a temperature of 40 °C, with a frequency of 1 Hz. Dynamic strain sweep tests are commonly used in materials science and engineering to characterize the viscoelastic properties of a material. The general purpose of dynamic strain sweep tests is to study the behavior of a material under cyclic loading at different strain amplitudes over a range of frequencies. Deformation was carried out in the range 1–20 μm, dividing this range into 5 equal segments, thus increasing the deformation within a cycle from 1 μm to 5 μm, to 10 μm, to 15 μm and reaching 20 μm, after which the force was removed and a cycle was completed. A similar cycle was performed for the three FeMnSi–2Ag samples. The difference consists in the deformation frequency, which was 1 Hz for the first sample (FeMnSi–2Ag(I)), 5 Hz for the second (FeMnSi–2Ag(II)) and 20 Hz for the third (FeMnSi–2Ag(III)). 

Immersion in Ringer’s biological solution (chemical composition [g/L]: NaCl—6.5; KCl—0.42; CaCl_2_—0.25; NaHCO_3_—0.20) at 37 °C was carried out with the pH of the solution recorded during the first 72 h. The corrosion rate based on mass loss was determined for the FeMnSi–Ag sample before and after applying the strain sweep. The samples before and after DMA-SS were subjected to an immersion corrosion resistance test. The in vitro tests involved immersion of hot-rolled FeMnSi–1Ag and FeMnSi–2Ag plates previously subjected to the DMA test and samples of the same alloys not subjected to the DMA test in biological Ringer’s solution for 14 days at 37 °C in a thermostatically controlled enclosure. Samples placed in Ringer’s solution (20 mL/cm^2^ ratio) were prepared by grinding on metallographic paper up to 5000 grit, cleaned in technical alcohol and weighed with an AS220 Partner analytical balance. After immersion, the samples were cleaned in an ultrasonic bath in technical alcohol for 60 min each. Weighing resulted in initial masses after immersion and after ultrasonic cleaning, which were used to calculate corrosion rates according to ASTM G31-03 using the formula (*w*: weight loss in [g], *A*: exposed area in [cm^2^], *t*: time in [h] and *ρ*: density in [g/cm^3^]) [53]:(8)$$CR=\frac{8.76\times 10^{4}\;W}{At\rho },$$

Fourier transform infrared spectroscopy (FTIR) was performed on a Bomem MB154S FT-IR spectrometer (Bomem, ABB group, Quebec city, Quebec, Canada) at an instrumental resolution of 4 cm^−1^. The samples were analyzed as powders which were obtained by scraping from the filter material. Each of the sample powders obtained was mixed in a mortar with KBr, and the mixture was pressed at a pressure of 100 atm to form pellets. We utilized a commercial scattering near-field microscope (neaSCOPE, attocube.com, accessed on 20 May 2023), equipped with a tuneable laser (MIRCAT, Daylight Solutions) to acquire monochromatic nano-IR images. The IR radiation, attenuated to a few mW power, was focused on a metallized tip (NCPt arrow, nanoandmore.com, accessed on 20 May 2023) using a parabolic mirror. The near-field optical interaction between the tip and sample was modulated by operating the atomic force microscopy (AFM) in tapping mode.

AFM was used to assess the surface morphology evolution of the samples before and after strain sweep process. Nanosurf EasyScan II (Liestal, Switzerland) was used to scan the surfaces (50 × 50 and 3 × 3 μm^2^) using a Si tip (PPP CTR10) in the contact mode.

The structure of the alloys and their condition before and after immersion in Ringer’s solution was investigated using scanning electron microscopy (SEM). VegaTescan LMH II (TESCAN, Brno-Kohoutovice, Czech Republic) equipment with cathode supply voltage of 30 kV, secondary electron detector and working distance of 15.5 mm was used. The chemical composition analysis of the alloys, their surfaces after immersion and the distribution of elements on the surface was performed with an X-ray energy detector (energy dispersive spectroscopy (EDS), Bruker, X-Flash 6-10, Billerica, MA, USA).

## 3. Results and Discussion

### 3.1. Microstructural (SEM) and Chemical (EDS and XRD) Analysis

In the as-cast state, both alloys exhibit a dendritic microstructure. The appearance of the microstructure changed after heat treatment at 1173 K (900 °C) for 30 min and water cooling. This revealed austenite grains with martensite plates inside that were several hundred micrometers in diameter (Figure 1a,b). No undissolved Si-rich precipitates can be observed after hot rolling treatment in the microstructure as in other FeMnSi SMA. Compared to FeMn alloy, the addition of Si resulted in a better recrystallization treatment, resulting in a more homogenized microstructure with an average grain size of 40 μm ± 5 µm (mean value and standard deviation) [54]. Similar to other results in the literature, silver separation was observed based on the very low solubility of silver in the solid solution of iron–manganese [55,56]. Besides silver dispersion, which could be beneficial for the degradation rate or antibacterial properties, in a few cases a high Ag agglomeration was obtained.

FeMnSi–1Ag and FeMnSi–2Ag have a complex microstructure consisting of γ-austenite (fcc structure) and ε-martensite (hcp structure) at room temperature. Small plate-like ε-martensite zones can be observed on the FeMnSi–2Ag alloy structure (Figure 1b) [54].

The chemical composition of the alloys is shown in Table 1 for areas larger than 1 mm^2^ in the first two rows and at three different selected points on the 90 nm diameter surface marked in Figure 1b. The average Ag content is close to 1 and 2 wt % in the alloys, and the large differences shown between the areas of the three selected points highlight the very low solubility of Ag in Fe, Mn or FeMn solid solution and the fact that they separately form compounds with a high percentage of Ag, such as points 2 and 3. In these areas the FeMn–Ag phase or pure Ag contaminated with FeMnSi solid solution is present at the surface.

The XRD plots of the alloys (Figure 2) confirm the observations in the SEM images (Figure 1) and indicate the presence of two phases: γ-austenite as the main one and ε-martensite. Note that it has been experimentally observed that a Mn content between 13 and 22 wt % in binary Fe–Mn alloys favors the presence of the ε-martensite phase, while a Mn content higher than 22 wt % favors the presence of the γ-austenite phase [57]. However, in the case of Fe–Mn–Si alloys, this observation is not as pronounced as expected due to the presence of 4–5 wt % Si in the alloy. These observations show that martensitic transformation γ → ε is much more enhanced in Fe26Mn5Si and Fe23Mn5Si alloys compared to Fe30Mn5Si. For alloys with higher Mn content, the elastic effect of the austenite matrix, when subjected to external stresses, has been highlighted. The effect of Si addition in FeMn alloys has been reported by Gavriljuk et al. [58] showing a lower stacking fault energy of the austenite phase in Fe–Mn–Si alloys with increasing Si content, which also favors the formation of ε-martensite).

The microstructural information can be correlated with the DSC curves in Figure 3, which illustrate reverse martensitic transformations during the heating process. According to the XRD results, the presence of both martensitic phases was indicated on the diffractograms and their reversion to γ-fcc austenite can be identified by the DSC results. The ε-martensite peaks are higher for the hot-rolled samples (Figure 2b) based on the presence of stress-induced martensite in the alloy after the deformation process. Also observed in the XRD results, the peaks at 58 and 80 (2θ) are also attributed to the ε-phase [58]. The other peaks are attributed to body centered cubic (BCC) α’-martensite together with retained FCC γ-austenite. The main martensitic transformation in Fe-based SMAs is based on deformation (stress-induced) and less on the thermal effect on cooling (thermally induced). XRD experiments at different sample temperatures were performed in order to establish the main structural components (−50, 0, 25 and 100 °C), Appendix A. All samples analyzed are mainly austenitic phase type. The crystal structure of γ-austenite is face-centered-cubic (FCC), Copper model cF4 [59]. The cell parameter has a bigger value of the pure γ-Fe (0.36 nm), as preconized paying respect to the fact that this phase solid solution of Mn, Si and less Ag in Fe. Other common phases found in this system are δ-ferrite in the sample at 25 °C and stress-induced ε-martensite observed in all cases except the sample chilled down to −50 °C [58]. The α’ phase like type crystallizes in a body-centered cubic structure, W-type—Im3m. On the other hand, ε-martensite exhibits hexagonal symmetry HCP, Mg-type—P63.

### 3.2. Differential Scanning Calorimetry (DSC) Analysis

Typically, for FeMn-based SMAs the martensitic phase transformation occurs over an extended temperature range and the exothermic peak, associated with martensite formation during cooling, is much less prominent compared to NiTi or CuZnAl materials. The corresponding A_50_ temperature (start of direct martensitic transformation, on cooling) is significantly reduced to 25 °C or below 0 after 1 or 2 heating–cooling cycles. The transformation continues during the cooling step beyond the measurement limit of 223 K (−50 °C). This behavior could be related to crystal lattice distortion due to the higher manganese content in the alloy [57]. Compared to the Drevet et al.’s [60] findings, the A_s_ point is lowered by the addition of Ag from 50 °C in the temperature range of 0–25 °C, which may favor the use of these materials in the medical field.

Since the direct martensitic transformations (γ → ε) were less intense, the corresponding XRD pattern in Figure 2 shows a better preserved large amount of γ-austenite phase. However, even though the martensitic transformation temperature decreased in the FeMnSi–2Ag alloy, the kinetics of the transformation reaction was delayed by the increased manganese content in the alloy. It is likely that these alloys with shape memory capabilities could be used as implant materials inside the human body.

### 3.3. Dynamic Mechanical Analysis (DMA)—Strain Sweep (SS) and Atomic Force Microscopy (AFM)

Figure 4 shows the variations of average values of E’ and tanδ with strain amplitude for all specimens investigated. During the dynamic strain sweep test (SS), the samples are subjected to sinusoidal deformation (cyclic loading) with varying amplitude while the frequency is held constant. The resulting stiffness response of the material is then measured as the main output, and this stress response can be decomposed into two components: the elastic component (representing the energy stored in the material) and the viscous component (representing the energy dissipated in the material). This test was designed to find the regions of linear material behavior. The experiments were performed at a temperature of 40 °C, close to body temperature. A standard frequency-dependent behavior of modules corresponds to a typical dependence: the higher the frequency, the higher the module. The tests were performed with different frequencies and time durations (at the frequency of 1 Hz for FeMnSi–1Ag alloy the test duration was 9.6 min, resulting in a number of cycles: 9.6 × 60 × 1 = 576, when repeated 5 times—2880 cycles, and for FeMnSi–2Ag: 3090 cycles, for 5 Hz: FeMnSi–2Ag: 16.5 × 60 × 5 = 4950 cycles, repeated 5 times—24,750 cycles, for 20 Hz—higher number of cycles per second: FeMnSi–2Ag: 26.7 × 60 × 20 = 32,040/cycle × 5 = 160,200 cycles for the samples tested). Depending on the application type (arm: traction, foot: compression), implants may be required at different frequencies and time durations, a day, a week, a year or over a lifetime (e.g., for a lower limb implant, the number of requirements may vary between 5000 and 15,000 steps per day, and for an average of 10,000 steps, 3,650,000 requirements per year). Biodegradable materials are always used with protective coatings for the first periods of immersion to achieve the desired mechanical properties and after 6–12 months of working the material may start to degrade. In this case, the material already performs several hundred thousand cycles or several million cycles depending on the application and the material will start to degrade at a certain stress state. By applying SS cycles, we induce a stress state similar to the application case and analyze the corrosion-resistance behavior of the material after this stress process (about 2 weeks of use on an active lower limb implant).

The addition of silver to the FeMnSi alloy does not influence much the dynamic behavior of the material (Figure 4a,b) with a similar variation of the storage modulus or tanδ coefficient with slightly small differences in values, 2 wt % Ag increasing the average tanδ value. For the low frequency, 1 Hz, the damping behavior of the material, tanδ, decreases with the repetition cycle, the fifth cycle, based on the increase of the storage modulus up to 0.01% strain when a plateau occurs by the end of the experiment. A higher damping capacity of an implant material may be useful for the integrity of the material in the first implantation period. For higher frequencies of 5 and 20 Hz, respectively, the internal friction of the material (damping capacity) increases with the stress cycle and frequency from 0.013 at 1 Hz (FeMnSi–1Ag) to 0.022 at 20 Hz (FeMnSi–2Ag) (Figure 4c,d). The first stress cycle of metallic material usually shows different and variable results depending on the misalignments, microcracks or voids in the material structure. The storage modulus at 1 Hz of the material increases with the degree of deformation at 0.01% deformation from 222,000 to 232,000 MPa and maintains its highest value until the end of the test. At the frequency of 5 Hz, the storage modulus initially increases with deformation, after which, starting at 0.0075% strain amplitude, relaxation occurs and the modulus values decrease to the initial values (Figure 4c). At 20 Hz, the storage modulus shows a continuous and gradual increase with increasing deformation and in this case no plateau can be observed.

The effects of DMA-SS on the material structure induced by strain sweep frequency were investigated using AFM equipment. Experiments were conducted on the central area of the specimen subjected to DMA-SS before and after dynamic mechanical stress. Figure 5 shows the 3D AFM scanned areas of the specimens before and after the DMA-SS tests.

All AFM images of the surface show areas of martensite plates. The AFM images show a refinement of the martensite plates with increasing DMA-SS-3PB frequency, the values shown in Table 2. Systematic measurements were performed to quantify the impact of the strain frequency values on the martensite plate dimensions. For this purpose, five characteristic groups of martensite plates and five parallel plates within each group were chosen.

In all vibration cases (1 Hz and 5 Hz), the profile of the samples shows a decrease in plate sizes down to the nanometer range. The change in plate size influences the shape memory properties [61,62] and will change the corrosion behavior of the material.

### 3.4. Evaluation of In Vitro Corrosion Behavior and pH Monitoring

The metal interacts with the liquid medium to release positive ions (M^n+^) while keeping the electrons (e^−^) on the metallic base. A protective coating of metal oxide (colored marks) is created on the surface as a result of reactions. The deposition of calcium phosphate on the metal oxide layer, which results from interactions with biological fluids, enables cells to cling to the surface and create tissues [63].

Two hot-rolled FeMnSi–2Ag samples, one without DMA-SS and the other with DMA-SS, were immersed in Ringer’s electrolyte solution at 37 °C for three days, and the pH of the solution was continuously monitored by recording the values minute by minute. Variations in pH are shown in Figure 6. Increased values are observed in both cases, with a higher rate for the sample without DMA-SS. It can also be noticed from the curves obtained by recording the pH that there is a greater variation of the pH for the solution in which the sample without DMA-SS was immersed, which confirms the structural change and the increase in the number of martensite plates by reducing their size observed by AFM. After 28 days of immersion in r-SBF solution, Li et al. [64] observed a boost in pH up to 7.8 for an iron structure with 80% porosity manufactured through direct metal printing.

Following equation (8), corrosion rates were determined for each mass gain/loss of hot-rolled FeMnSi–2Ag samples subjected to pH monitoring for 72 h (Table 3). Sample areas were determined: FeMnSi–2Ag = 2.8 cm^2^ (sample without DMA-SS) and FeMnSi–2Ag = 2.6 cm^2^ (sample with DMA-SS). The corrosion rate results are shown in Table 3. A higher corrosion rate was obtained for the stressed sample (DMA-SS).

Table 4 presents the degree of oxidation and the main elements identified on the immersed samples according to EDS analysis. Although the mass loss is higher for the stressed material (DMA-SS), the degree of oxidation is not higher, which means that the degradation of the material occurs faster.

Considering the standard reduction potentials of Fe of −0.44 V and Mn of −1.18 V [61], the Fe–Mn alloy exhibits a more active corrosion potential due to the overall electron potential reduction of iron due to the addition of Mn [65,66,67,68]. Regarding the phases found in Fe–Mn alloys microstructure, α-ferrite (BCC) shows a better corrosion rate than ε-martensite (HCP) and γ-ferrite (FCC) [69]. The galvanic corrosion occurrence as a result of the association of these phases and the volume in which they are found in the material [70] would increase the corrosion rate of the alloy [52,65]. XRD results on the oxidized surface after immersion highlight the presence of γ and ε phases on the surface alongside iron oxides, manganese oxide, iron hydroxides and carbonates [71], Appendix A.

The mass loss (five samples were immersed and the average values are given in Table 5) recorded by immersion tests was applied to determine the material corrosion rate using Equation (8). An increase in corrosion rate is observed after DMA-SS stressing of the samples and a higher corrosion rate was obtained for the sample vibrated at a higher rate, i.e., 20 Hz.

The chemical element distribution (Figure 7) shows homogeneity on the surface after the immersion period, especially for the stressed FeMnSi–2Ag sample at 20 Hz (Figure 7f). Carbonate compounds appear on the metal surface after immersion among chlorides and phosphates.

SEM surface micrographs (at different scales of 50×, 500× and 2500×) were taken on the stressed DMA-SS samples before and after immersion in Ringer’s solution for 14 days at 37 °C. The surface of the materials is generally clean with no large cracks or pores, oxide compounds or other structural defects that may cause corrosion acceleration. After the immersion period, the surfaces in Figure 8d–l indicate generalized corrosion with intergranular corrosion for all samples from macro (1 mm) to micro (20 μm) scale.

For all cases, one phase appears to be more corroded (Figure 8e,h,k) presumably the austenite phase which is less resistant. At this stage, the undissolved phase showed corrosion pits of 1–2 μm in diameter. The materials show different stages of corrosion from macro to micro scale. The mechanical integrity of the materials was not affected at the surface; no major cracks were observed at the edges of the samples or deep pits at the surface. All samples show non-metallic compounds interconnected with the metal surface. The sample surface stressed at 5 Hz is cleaner and homogeneously corroded. At the micro scale, non-metallic compounds smaller than 1 μm were observed (Figure 8f) and areas not completely corroded with martensitic plates were identified (Figure 8l), confirming the higher strength of the martensite phase compared to the austenite phase.

### 3.5. Fourier Transform Infrared Spectroscopy (FTIR) and Nano-FTIR Analysis

FTIR and nano-FTIR techniques were used to identify and characterize the FeMnSi–Ag corroded surface [72,73]. During scraping, fibers were also removed from the filter material, which are observed on the FTIR spectra in Figure 9. The changes in the fibers’ spectra are attributed to the metals in the powder composition. A shift of the vibrational peak of the alcoholic hydroxyl groups (^−^OH) can be observed from 3437 cm^−1^ [74] in the fiber sample to about 3300 cm^−1^ in the spectra of all other samples (FeMnSi–Ag, FeMnSi). The shift can be assigned to the H-bonds and van der Waals interactions between the fiber molecules and the metallic composition of each sample. The spectra also showed the spherical structure of the magnetic particles in the powders of all the six samples by the shape of the baseline, which drops at the left side due to the Mie scattering effect [75,76]. The scattering effect is more pronounced in metallic samples and almost undetected in sample fiber. High water adsorption of filter fiber is still evident on the corrosion products spectra by the noisy aspect they exhibit in the ranges of 4000–3700 cm^−1^ and 1695–1527 cm^−1^, respectively. The affinity toward water decreases in fiber samples where the broad peaks from 3300 cm^−1^ are maintained, indicating the existence of intra- and intermolecular hydrogen bonds at the fiber level. In the other samples, the OH peaks narrow and move toward 3427, signifying that the H bonds are less.

The band at 1622 cm^−1^ corresponded to the vibration of aromatic groups (e.g., C=C) and carboxyl (C=O) [77].

The skeletal vibrations of the fiber denoted by the peaks at 1023 cm^−1^ exhibit changes in shape and left shift in samples, indicating interactions between the filter fibers and the particles of metals. The very sharp peak at 1023 cm^−1^ in the FeMnSi–Ag spectrum can be assigned to residual NaHCO_3_ [76] from the Ringer’s solution.

In addition to the fiber signal, part of the bands are proper new bands at ν = 1428, 1162, 876, 697, and 566 cm^−1^ observed in the FTIR spectra of FeMnSi and FeMnSi–Ag compounds. The peaks from 1428 and 1162 cm^−1^ may correspond to the Fe–OH and Mn–OH compounds (Equation (4) and Table 4) [78,79]. The peak band observed at 876 cm^−1^ may be related to the Fe–OH vibrations of α-FeOOH. Moreover, the absorption bands at 697 and 566 cm^−1^ could be related to α-Fe_2_O_3_ [76]. Meanwhile, the small peaks observed at 520 and 720 cm^−1^ can be ascribed to the Mn–O vibrations. No different visible bands were identified between FeMnSi and FeMnSi–Ag degradation compounds, probably because of the small amounts of Ag from the sample and high resistance to corrosion of the compounds formed with Ag.

In Figure 10, Figure 11 and Figure 12, a new nanoscale analysis technique was used to show the oxide compounds identified on FeMnSi–Ag alloys after atmospheric exposure and immersion in Ringer’s solution for 24 h. No surface cleaning process was performed to observe the compounds on the surface and to characterize the beginning of the degradation process of this nanoscale and early material. During the sample scanning process, we employed a Michelson interferometer in a pseudoheterodyne detection scheme to detect and analyze the tip-scattered light and eliminate far-field background scattering [80]. This allowed us to simultaneously record AFM topography, mechanical phase, IR absorption and reflectivity properties. The spatial resolution primarily depended on the probing tip radius (approximately 20 nm), while the optical contrasts were influenced by the local optical properties of the material beneath the tip and the IR illumination wavelength. In nano-FTIR spectroscopy mode, the neaSCOPE system utilized coherent broadband mid-infrared illumination from a DFG laser source (Toptica). The back-scattered light was detected using an asymmetric Michelson interferometer [81].

The scanned surface is shown in Figure 10, left side, at a scale of 2 μm, and the profile and compounds observed on the surface were smaller than 100 nm. Corrosion sites are revealed using different types of illumination of the surface profile (AFM scanning by tapping with laser light focused on a sharp AFM tip, all-optical interferometric detection recovers both the amplitude and the phase of the light scattered by the tip, providing complete information about the complex optical properties of the sample, (e.g., absorption and reflectivity)).

The corroded sample had mainly two types of zones. One with a relatively thin layer and the other with a relatively thick layer, as shown in the images in Figure 10 and Figure 11. Several compounds and areas on the surface, all at nanoscale, were selected for FTIR investigations to identify the compound types.

Apart from the thick layer spectra, a broad and strong band can be observed at about 1100 cm^−1^ (marked with a blue circle) similar to that of FeMnSi sample and therefore not related to Ag addition. For 1100 cm^−1^ there are three possibilities: one can be attributed to the presence of SiO_2_, the second to a carbonate compound or to the presence of a C–O bond.

At the thin layer level, a particle with a strong and broad band (similar to phonon absorption of oxides) at about 800 cm^−1^ (orange spectrum) can be observed, this fact indicates the presence of a common inorganic ion such as CO_3_^2−^, which is confirmed by EDS analysis (the results shown in Table 4). At the same time, Veneranda et al., show in their work the goethite compound (α-FeOOH) at 798 cm^−1^, which is more likely to cover the entire surface of the material [82]. In the thick layer, two particles are observed with a clear peak at about 1590 cm^−1^. Note that the near-field amplitude of these particles is very small and therefore the spectra are quite noisy. Su et al., identified at the 1590 cm^−1^ peak the presence of COO^-^ which can be attributed to the carboxyl group stretching [83].

The near-field phase image at 1085–1121 cm^−1^ shows more contrast again, similar to the FeMnSi sample, indicating that the broad band at about 1100 cm^−1^ is heterogeneously distributed on the surface. Interestingly, this image also shows particles (marked with red circles) that absorb with comparable strength. The near-field phase image at 910 cm^−1^ reveals the particle that produced the broad phonon-like absorption centered at 800 cm^−1^. The peak at approximately 910 cm^−1^ which is clearly seen in this spectrum can be attributed to the origin of the signal from Si–O–H stretching [84,85].

## 4. Conclusions

The presented results confirm the possibility of obtaining a homogeneous and smart FeMnSi–Ag alloy with shape memory using the levitation furnace. The influence of an external stress, achieved by using a dynamic mechanical amplitude sweep, on the microstructure, and corrosion-resistance behavior was determined. The main corrosion products were identified and a degradation rate was established. The experimental results led to the following conclusions:-the levitation furnace is a proper solution for manufacturing FeMnSi–Ag alloys with different amounts of Ag added (1, 2 wt %).-the materials exhibit excellent structural and chemical homogeneity, with Ag spreading within the structure (γ-austenite and ε-martensite phases were identified by XRD).-the DSC results validate the possibility of medical use of a functional FeMnSi-based material with an A_50_ temperature between 0 and 50 °C.-Preceding the start of the in vitro degradation process, DMA_SS was used to vibrate the material for daily use as implant material.-AFM studies have shown that the alloy has undergone a structural change due to stress-sweeping effects, which has also led to an increase in the amount of ε-martensite plates, a phase less resistant to corrosion than the γ-austenite phase.-nano-FTIR spectroscopy and imaging can currently provide spectral statistics of externally corroded metal with a spatial resolution of 20 nm.-Nano-scale compounds were identified on the metallic surface after material immersion, confirming that the corrosion process evolves from nano to macro scale.

## Figures and Tables

**Figure 1 jfb-14-00377-f001:**
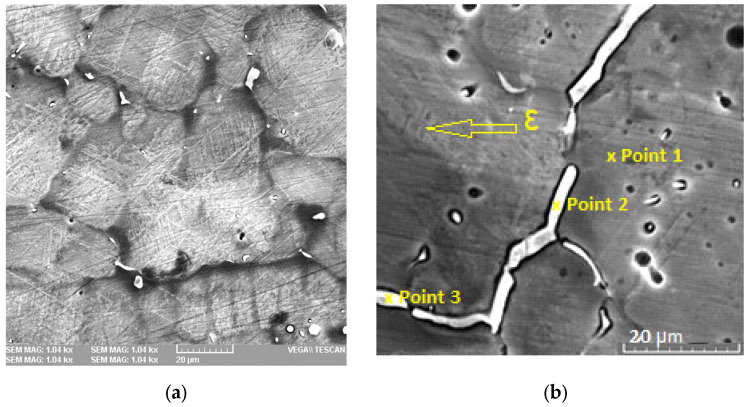
SEM micrographs of the experimental alloys: (**a**) FeMnSi–1Ag; (**b**) FeMnSi–2Ag.

**Figure 2 jfb-14-00377-f002:**
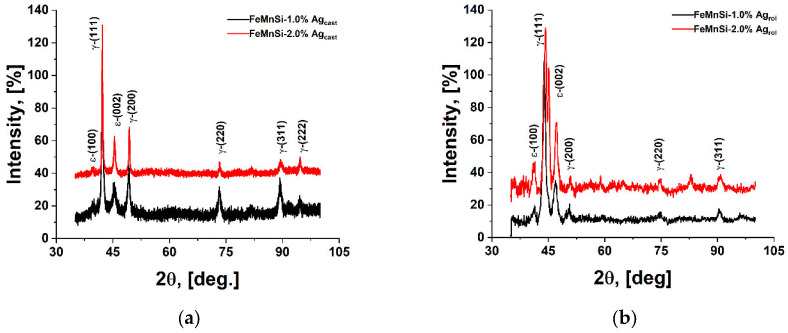
XRD patterns of FeMnSi–1Ag and FeMnSi–2Ag samples: (**a**) cast and (**b**) hot rolled.

**Figure 3 jfb-14-00377-f003:**
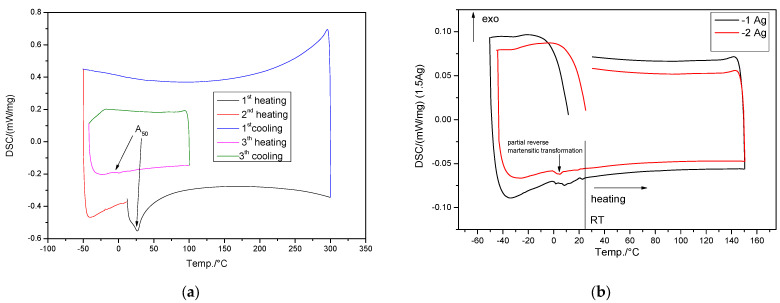
DSC diagrams for the hot-rolled FeMnSi–1Ag and FeMnSi–2Ag samples: (**a**) FeMnSi–2Ag for two heating/cooling cycles and (**b**) cooling–heating cycles of FeMnSi–1Ag and FeMnSi–2Ag SMAs, respectively.

**Figure 4 jfb-14-00377-f004:**
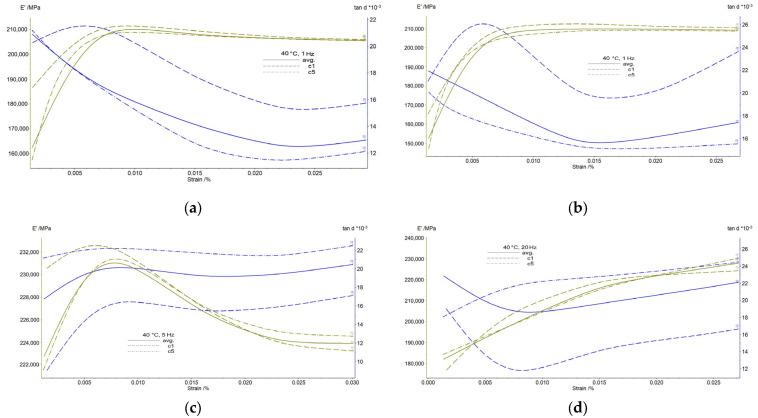
DMA_SS results for the hot rolled plates (25 × 4 × 1 mm): (**a**) FeMnSi–1Ag (40 ℃, 1 Hz); (**b**) FeMnSi–2Ag(I) (40 ℃, 1 Hz); (**c**) FeMnSi–2Ag(II) (40 ℃, 5 Hz); (**d**) FeMnSi–2Ag(III) (40 ℃, 20 Hz).

**Figure 5 jfb-14-00377-f005:**
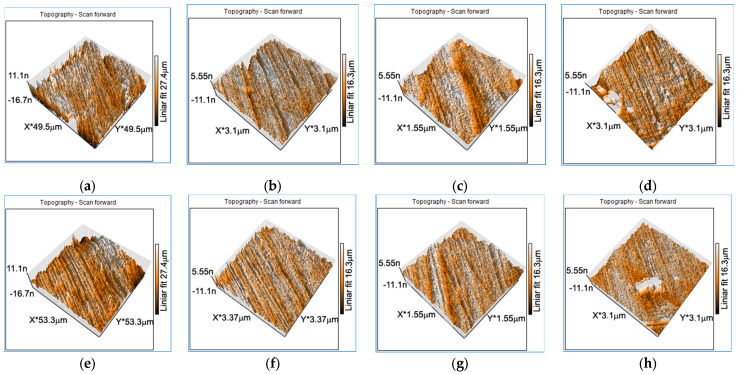
AFM images (3D topography) of the hot rolled plates before DMA-SS testing: (**a**) FeMnSi–1Ag; (**b**) FeMnSi–2Ag (I); (**c**) FeMnSi–2Ag (II); (**d**) FeMnSi–2Ag (III); and after DMA-SS testing: (**e**) FeMnSi–1Ag (1 Hz); (**f**) FeMnSi–2Ag (I) (1 Hz); (**g**) FeMnSi–2Ag (II) (5 Hz); (**h**) FeMnSi–2Ag (III) (20 Hz).

**Figure 6 jfb-14-00377-f006:**
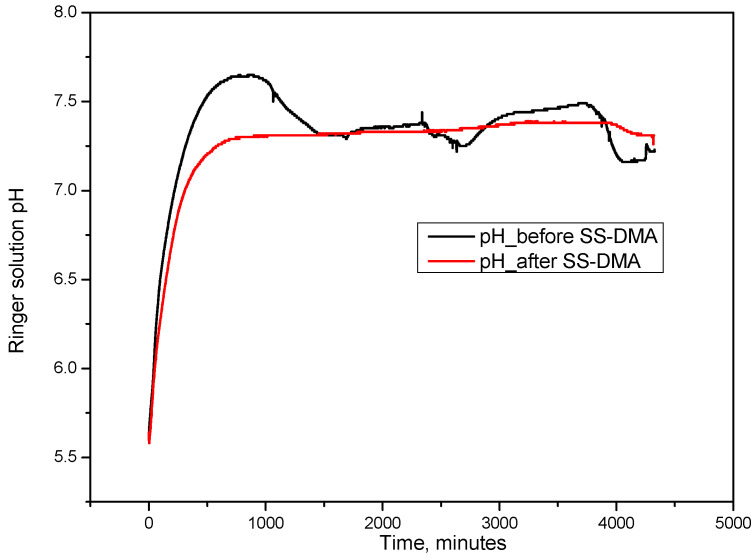
The pH variation recorded for 72 h of immersion in Ringer’s solution for the hot rolled FeMnSi–2Ag samples, without and with DMA-SS.

**Figure 7 jfb-14-00377-f007:**
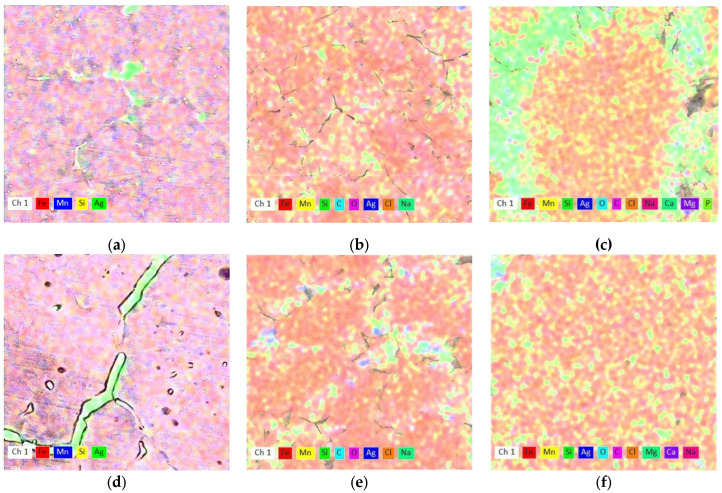
Chemical element distribution of the hot-rolled samples for the FeMnSi–1Ag alloy: (**a**) initial state; (**b**) after 14 days immersion without supplementary solicitation applied; (**c**) with DMA-SS; and for the FeMnSi–2Ag alloy: (**d**) initial state; (**e**) after 14 days immersion without supplementary solicitation applied; (**f**) after 14 days immersion with DMA-SS.

**Figure 8 jfb-14-00377-f008:**
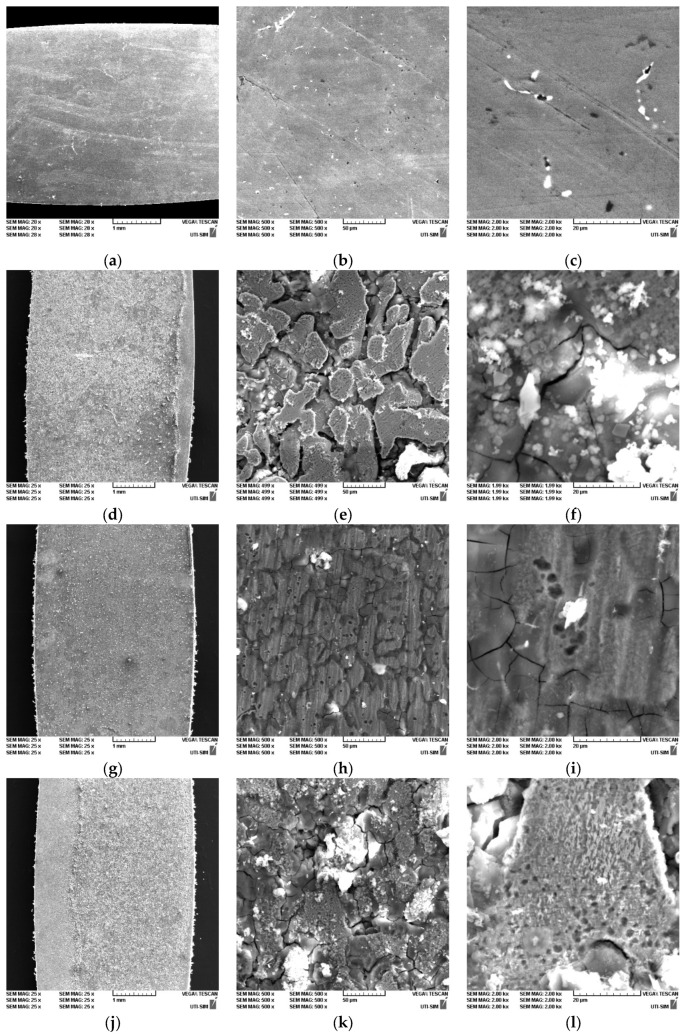
SEM images of the hot rolled plates, before: (**a**–**c**) FeMnSi–2Ag and after DMA-SS: (**d**–**f**) FeMnSi–2Ag (1 Hz); (**g**–**i**) FeMnSi–2Ag (5 Hz); (**j**–**l**) FeMnSi–2Ag (20 Hz).

**Figure 9 jfb-14-00377-f009:**
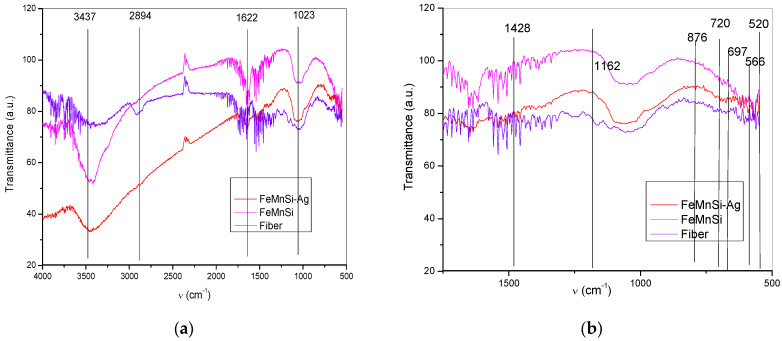
FTIR spectra of the metallic powder: textile Fiber, FeMnSi and FeMnSi–Ag: (**a**) main spectra and (**b**) detail of 500–1750 cm^−1^ region.

**Figure 10 jfb-14-00377-f010:**
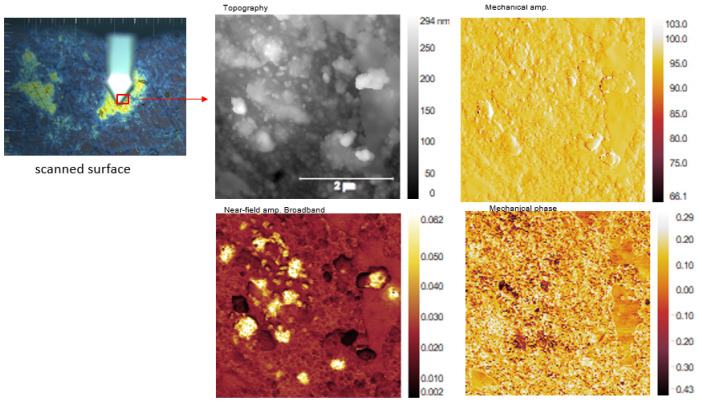
Optical and AFM images of the FeMnSiAg surface as corroded sample.

**Figure 11 jfb-14-00377-f011:**
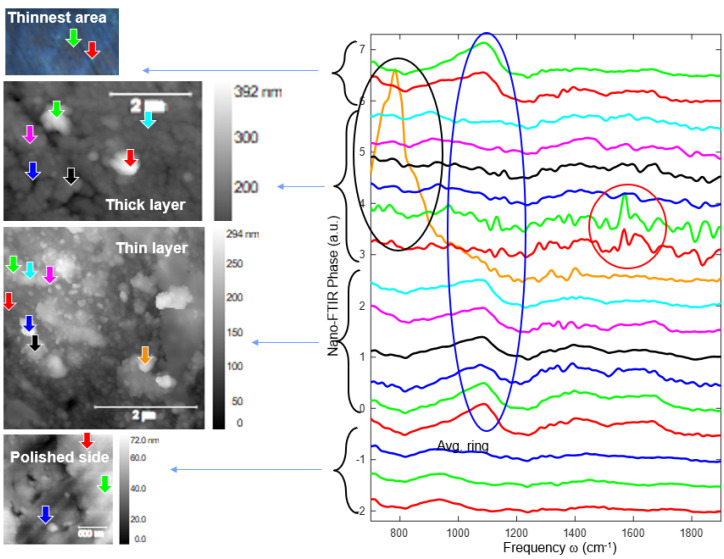
Scanned surface and the dedicated FTIR spectra of the particles observed on the FeMnSi–Ag alloy.

**Figure 12 jfb-14-00377-f012:**
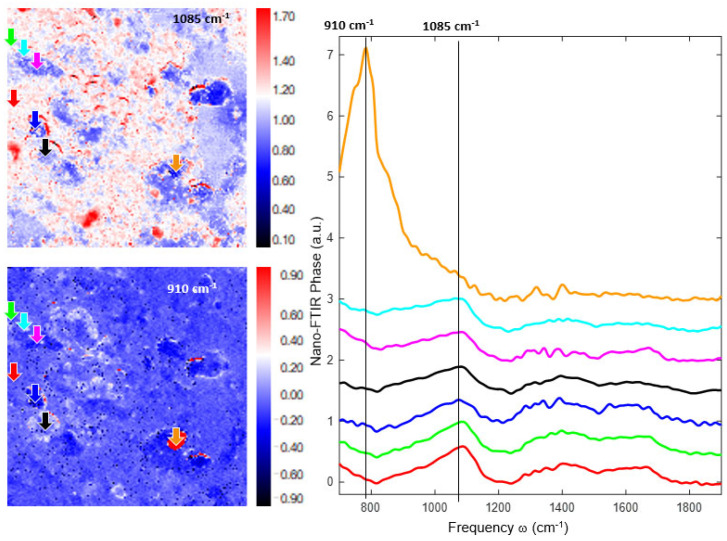
s-SNOM images and nano-FTIR spectra fit together for FeMnSiAg surface.

**Table 1 jfb-14-00377-t001:** The initial chemical composition of the FeMnSi–1Ag and FeMnSi–2Ag alloys.

Alloy	Fe		Mn		Si		Ag	
	Wt %	At %	Wt %	At %	Wt %	At %	Wt %	At %
FeMnSi–1Ag	64.1	61.48	30.84	30.08	4.19	7.99	0.9	0.45
FeMnSi–2Ag	63.32	60.82	30.27	29.56	4.56	8.71	1.86	0.92
Point 1	69.45	67.00	27.29	26.76	3.26	6.24	0.01	0.04
Point 2	37.16	38.15	36.78	38.39	6.37	13.00	19.68	10.46
Point 3	55.57	58.17	24.75	26.25	3.28	6.82	16.22	8.76
Detector err. %	1.45		0.75		0.25		0.6	

St. dev.: Fe: ±1.5; Mn: ±1; Si: ±0.2, Ag: ±0.2.

**Table 2 jfb-14-00377-t002:** AFM measurement results of martensite plates (average of 5 determinations on 5 plates for each sample) and standard deviation percentage (st. dev.%) for FeMnSi–1Ag and 3 x FeMnSi–2Ag samples, before and after DMA-SS testing.

Sample	Dimension	Average before DMA-SS (nm)	St. dev.%	Average after DMA-SS (nm)	St. dev. %
FeMnSi–1Ag (1Hz)	l	334.8	86.9	41.8	6.5
	h	204.8	40.1	30.4	5.5
FeMnSi–2Ag(I) (1Hz)	l	107.8	16.1	59.6	25.1
	h	85	9.5	45.6	18.03
FeMnSi–2Ag(II) (5Hz)	l	92	20.2	58	13.1
	h	105.4	56.8	44.6	9.3
FeMnSi–2Ag(III) (20Hz)	l	81.2	16.6	54.8	4.9
	h	71.4	8.9	50.4	8.2

**Table 3 jfb-14-00377-t003:** Corrosion rate according to mass gain/loss for FeMnSi–2Ag hot rolled samples subjected to pH monitoring for 72 h.

Hot-Rolled Sample	FeMnSi–2Ag without DMA-SS	FeMnSi–2Ag (20 Hz) with DMA-SS
Initial mass (mg)	796.7	725.3
Mass after immersion (mg)	797.7 (+1.0)	725.5 (+0.2)
Mass after ultrasound (mg)	794.4 (−2.3)	722.5 (−2.8)
**Corrosion rate (μm/year)**	**134.15**	**175.87**

**Table 4 jfb-14-00377-t004:** EDS results after the 14 days immersion tests in Ringer’s solution and ultrasound cleaning for the FeMnSi–1Ag and FeMnSi–2Ag hot-rolled samples.

Chemical Elements/Sample	Fe		Mn		Si		Ag		O		C		Na		Cl		Ca	
	Wt %	At %	Wt %	At %	Wt %	At %	Wt %	At %	Wt %	At %	Wt %	At %	Wt %	At %	Wt %	At %	Wt %	At %
FeMnSi–1Ag	36.76	16.5	9.35	4.27	9.16	8.18	2.17	0.5	34.67	54.32	7.69	16.04	0.12	0.13	0.09	0.06	-	-
FeMnSi–2Ag	42.92	21.6	16.06	8.22	5.56	5.56	0.61	0.16	26.43	46.43	7.2	16.86	0.46	0.56	0.76	0.6	-	-
FeMnSi–1Ag (1 Hz)	42.34	20.54	14.63	7.21	4.05	3.9	0.57	0.14	30.7	51.97	6.88	15.52	0.14	0.16	0.6	0.46	0.02	0.01
FeMnSi–2Ag(1 Hz)	40.6	19.8	14.83	7.35	3.94	3.82	1.68	0.42	32.58	55.45	5.38	12.19	0.40	0.48	0.42	0.32	0.05	0.03
FeMnSi–2Ag(5 Hz)	44.97	24.02	19.17	10.41	5.62	5.97	0.36	0.1	21.02	39.19	7.6	18.87	0.83	1.08	0.21	0.18	-	-
FeMnSi–2Ag (20 Hz)	43.35	22.1	15.75	8.16	5.86	5.95	0.89	0.24	26.96	47.98	6.21	14.72	0.09	0.12	0.63	0.51	0.1	0.1
EDS err. %	0.9		0.1		0.1		0.05		1.9		0.7		0.2		0.2		0.1	

StDev: Fe: ±0.9, Mn: ±0.7, Si: ±0.22, Ag: ±0.4, O: ±0.2, C: ±0.1, Na: ±0.1, Cl: ±0.1, Ca: ±0.1.

**Table 5 jfb-14-00377-t005:** Corrosion rates (CR) determined by each mass gain/loss of the FeMnSi–1Ag and FeMnSi–2Ag hot-rolled samples without DMA-SS previously applied and samples with DMA-SS testing, subjected to immersion tests and subsequent cleaning in the ultrasonic bath (immersion time was for 14 days in Ringer’s solution.

Sample	Samples without DMA-SS		Samples with DMA-SS			
	FeMnSi–1Ag	FeMnSi–2Ag	FeMnSi–1Ag (1 Hz)	FeMnSi–2Ag (1 Hz)	FeMnSi–2Ag (5 Hz)	FeMnSi–2Ag (20 Hz)
Initial mass (mg)	882.5	826.4	692.9	644.7	711.7	748.1
Mass after immersion (mg)	876.7 (−5.8)	843.1 (+16,7)	699.4 (+6.5)	646.2 (+1.5)	708 (−3.7)	760.3 (+12.2)
Mass after ultrasound (mg)	876 (−6.5)	818.5 (−7.9)	686.8 (−6.1)	636.3 (−8.4)	704.2 (−7.5)	737.7 (−10.4)
**Corrosion rate (μm/y)**	**64**	**79**	**82**	**113**	**101**	**140**

Samples areas without DMA-SS were calculated: FeMnSi–1Ag = 3.3 cm^2^ and FeMnSi–2Ag= 3.1 cm^2^; samples areas with DMA-SS were calculated: FeMnSi–1Ag, FeMnSi–2Ag(I), FeMnSi–2Ag(II), FeMnSi–2Ag(III) =2.6 cm^2^.

## Data Availability

Data is contained within the article or Appendix A.

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
