# Peer review of "Influence of Dynamic Strain Sweep on the Degradation Behavior of FeMnSi–Ag Shape Memory Alloys"

_jfb, 2023, doi:10.3390/jfb14070377_

Round 1

Reviewer 1 Report

The work comprehensively investigates the microstructure, dynamic mechanical behavior, and in vitro corrosion behavior of FeMnSi-Ag using multiple characterization techniques. This study significantly contributes to expanding the understanding of FeMnSi shape memory alloys in the field of biomedical materials. Thus, I suggest that it can be accepted for publication after a major revision. There are also some questions needed to be clarified as follows:

1.     It is suggested to provide a detailed explanation of the reasons behind conducting the dynamic strain sweep for FeMnSi-Ag in biomedical applications. Presenting the reasons and significance in the introduction section will enhance the reader's understanding of the work being presented.

2.     It is recommended to provide information on the surface preparation method for the samples before conducting X-ray diffraction (XRD), dynamic mechanical analysis (DMA), and differential scanning calorimetry (DSC) analysis. The surface condition of the samples can significantly impact their mechanical behavior during dynamic strain sweep experiments and the characterization results following immersion tests. However, the manuscript does not mention the sample preparation method prior to conducting the various characterizations.

3.     Please improve the annotation of phase transition points in Figure 3. Specifically, provide clear and accurate labels indicating the location of martensitic phase transition, if present, during the cooling process. Additionally, if there is no martensitic phase transition during cooling, explain the presence of an austenitic phase transition during the third heating cycle. Clear and concise labels will enhance the understanding of the phase transitions depicted in the figure 3.

4.     Supplemental Data:It is recommended to include in the supplementary materials the in-situ XRD results at temperatures of -50°C, 0°C, 25°C, and 100°C. By comparing the relative content of the martensite and the austenite phases at different temperatures, the authors can demonstrate the reduction in the austenite transformation temperature to the range of 0-25°C.

5.     Supplemental Data: The authors are advised to provide additional data on the shape memory effect of FeMnSi-Ag in supplementary materials,such as strain recovery rate. This will enable them to illustrate the influence of the silver element on the shape recovery rate of the FeMnSi-Ag shape memory alloy. Furthermore, these data will provide further support for the conclusions drawn from the DSC test results.

6.     Supplemental Data: XRD Analysis of Corrosion Products Please include XRD data depicting the composition of the corrosion products formed on the material's surface after immersion. This additional information will serve to clarify the constituents of the corrosion products resulting from in vitro immersion.

Author Response

Thank you very much for all your pertinent and excellent suggestions. 

Reviewer 2 Report

The paper studies the FeMnSi alloy, modified with Ag, as shape memory alloys for medical application. Dynamic mechanical solicitation, atomic force microscopy, dynamic mechanical analysis, SEM analysis, differential scanning calorimetry and Fourier transform infrared spectroscopy were performed to characterize the degradation and the influence of Ag. The paper is written well, but the structure is not optimal: theory is provided in results and rather moved to the front, and a discussion is missing (or part as the results, but not titled as this). However, one can see the confidence in conducting this study, the motivation is very clear and the results will be of value to the community.

See following comments and suggestions:

Title:

- ok

Abstract:

- rather long, can be shorten by not stretching theory and methods too much, rather results!

- there is no need for abbreviations in the abstract, rather start in the main text

Introduction:

- the paper starts with references, which do not refer to recent papers – please add some more recent to 1, 2, 3, 4…

- page 2, line 47: please add “design of potential medical implant”

- page 2, line 53: here “Shape memory alloys (SMA)…”

- page 3, line 111: you mention later, that Ringer is an aggressive aqueous medium, please add in your motivation, why you are using an aggressive medium

Experimental Materials and analytical techniques

- revision of headline in “Materials and Methods” is recommended

Experimental results

- headline needs to be changed in “Results and Discussion” – otherwise you have no discussion pointed out (there is no extra paragraph on discussion)

- page 4, line 182: “an average grain size of about 40 µm” needs to be provided more specific: 40 µm ± xx µm (mean value and standard deviation)

- page 7, line 241: please remove the distance between figure and figure caption

- page 7, lines 264 to 271: the arrow should be exchange to text and numbers for step per day need commas instead of spaces

- figure 5: please enlarge the labels, very difficult to read

- page 10: here the miss-structure is very clear – line 321 to 331 with formula (1) belongs to Methods,

lines 333 to 348 and 355 to 375 should be rather part of the Introduction, it is pure theory

- figure 6b is not described within the text and also belongs to the theoretical part (not your results)

Conclusion

- could be a bit more precise, a paper with a variety of results should be summarized in more detail

Author Response

(The authors gave the same response as above.)

Round 2

Reviewer 2 Report

Thank you for addressing most of the suggestions.

Please check new reference 6: it is in you comments, but not in the revised paper - GÄ…sior, G.; SzczepaÅ„ski, J.; Radtke, A. Biodegradable Iron-Based Materials—What Was Done and What More Can Be Done? Materials 2021, 14, 3381.

Very critical is still the issue with the Ringer solution. Now you have provided the following sentence: "In vitro experiments are performed by immersing samples in physiological solutions, which are aggressive aqueous media (e.g. Ringer's solution, Hank’s solution, Simulated Body Fluid (SBF))" and refered to mostly review papers: [4, 6, 10, 23].

My question was on your motivation to use Ringer in the first place and not a more suitable solution like HBSS or SBF. The senctence needs to be revised: HBSS and SBF are not classed as aggresive solutions. Instead of refering to review papers, you should provide resreach papers, which apply Ringer. Ringer is the most aggressive one and there is still the question, why you used this one.

The conclusion needs to have an "introduction" paragraph instead of jumping to bullet points - a short summary of your experimental and motovation is recommended for the sart of the conclusion.

Author Response

Thank you very much for your effort and support to improve the work.

The answers to your suggestions are presented below and we hope they will be suitable for them. 

Please check new reference 6: it is in you comments, but not in the revised paper - GÄ…sior, G.; SzczepaÅ„ski, J.; Radtke, A. Biodegradable Iron-Based Materials—What Was Done and What More Can Be Done? Materials 2021, 14, 3381.

Response: Thank you , we add the reference 6 on the final version of the paper

Very critical is still the issue with the Ringer solution. Now you have provided the following sentence: "In vitro experiments are performed by immersing samples in physiological solutions, which are aggressive aqueous media (e.g. Ringer's solution, Hank’s solution, Simulated Body Fluid (SBF))" and referred to mostly review papers: [4, 6, 10, 23].

My question was on your motivation to use Ringer in the first place and not a more suitable solution like HBSS or SBF. The senctence needs to be revised: HBSS and SBF are not classed as aggresive solutions. Instead of refering to review papers, you should provide resreach papers, which apply Ringer. Ringer is the most aggressive one and there is still the question, why you used this one.

Response:

Most of the results presented in the literature for FeMn-based biodegradable alloys are in SBF, HBSS, or Dulbecco solutions (references: [38], [39], [40], [41], [42]), and our intention was to observe the behavior of these alloys (FeMnSi-Ag) in Ringer's solution, considered more aggressive: the corrosivity of the solution media (from least to most corrosive) is given by PBS < SBF < Ringers scale (Mohd Amin Farhan Zaludin, 2019). These experiments are part of a PhD thesis, and experiments with different solutions in addition to Ringer's solution are proposed. The interest in Ringer is given by the aggressiveness of this solution and the possibility of forming large corrosion products.

We modify the paper text: In vitro experiments are performed by immersing samples in physiological solutions, which are more or less aggressive aqueous media (e.g., Ringer's solution, Hank’s solution, Simulated Body Fluid (SBF)) [4, 6, 10, 23].

Also we add the follow paragraph:

Corrosion mechanism of titanium dioxide nanotubes in Ringer's solution was studied by Zarebidaki [43] and Chelliah study the evaluation of electrochemical impedance and biocorrosion characteristics of as-cast and T4 heat treated AZ91 Mg-alloys in Ringer's solution [44].

[43] Zarebidaki, A., Mofidi, S.H.H., Nodezh, A.S., Corrosion mechanism of titanium dioxide nanotubes in Ringer's solution, Mat. Today Comm., 2021, 29, 102943.

[44] Chelliah, N.M., Padaikathan, P., Kumar R., Evaluation of electrochemical impedance and biocorrosion characteristics of as-cast and T4 heat treated AZ91 Mg-alloys in Ringer's solution, J. of Magn. and All., 2019, 7, 1, 134-143.

The conclusion needs to have an "introduction" paragraph instead of jumping to bullet points - a short summary of your experimental and motivation is recommended for the start of the conclusion.

Response: Thank you for this suggestion we modify the Conclusions Section and add in text:

The presented results confirm the possibility of obtaining a homogeneous and smart FeMnSi-Ag alloy with shape memory using the levitation furnace. The influence of an external stress, achieved by using a dynamic mechanical amplitude sweep, on the microstructure and corrosion resistance behavior was determined. The main corrosion products were identified, and a degradation rate was established.